# The Synthesis and Antibacterial Properties of Pillar[5]arene with Streptocide Fragments

**DOI:** 10.3390/pharmaceutics15122660

**Published:** 2023-11-23

**Authors:** Evgenia Subakaeva, Pavel Zelenikhin, Evgenia Sokolova, Arina Pergat, Yulia Aleksandrova, Dmitriy Shurpik, Ivan Stoikov

**Affiliations:** 1Institute of Fundamental Medicine and Biology, Kazan Federal University, Kremlevskaya, 18, 420008 Kazan, Russia; zs_zs97@mail.ru (E.S.); zhenya_mic@mail.ru (E.S.); 2A.M. Butlerov Chemical Institute, Kazan Federal University, Kremlevskaya, 29, 420008 Kazan, Russia; pergatarina@yandex.ru (A.P.); a.julia.1996@mail.ru (Y.A.); dnshurpik@mail.ru (D.S.)

**Keywords:** pillararene, sulfonamide, antimicrobial activity, antibiofilm activity, mutagenicity, toxicity

## Abstract

The growing problem of bacterial resistance to antimicrobials actualizes the development of new approaches to solve this challenge. Supramolecular chemistry tools can overcome the limited bacterial resistance and side effects of classical sulfonamides that hinder their use in therapy. Here, we synthesized a number of pillar[5]arenes functionalized with different substituents, determined their ability to self-association using DLS, and characterized antimicrobial properties against *S. typhimurium*, *K. pneumoniae*, *P. aeruginosa*, *S. epidermidis*, *S. aureus* via a resazurin test. Biofilm prevention concentration was calculated for an agent with established antimicrobial activity by the crystal–violet staining method. We evaluated the mutagenicity of the macrocycle using the Ames test and its ability to affect the viability of A549 and LEK cells in the MTT-test. It was shown that macrocycle functionalized with sulfonamide residues exhibited antimicrobial activity an order higher than pure streptocide and also revealed the ability to prevent biofilm formation of *S. aureus* and *P. aeruginosa*. The compound did not show mutagenic activity and exhibited low toxicity to eukaryotic cells. The obtained results allow considering modification of the macrocyclic platforms with classic antimicrobials as an opportunity to give them a “second life” and return to practice with improved properties.

## 1. Introduction

Antimicrobials created prior to the antibiotic era are not forgotten. These include sulfonamides synthesized at the beginning of 20th century [1]. Sulfonamides-derived drugs are still used in medical and veterinary practice due to their systemic and local action [2,3]. Sulfonamides have been shown to inhibit the growth of Gram-positive and Gram-negative bacteria, as well as some protozoa [4]. However, nearly a century-long history of the application of such antimicrobials is marred by the ability of pathogens to rapidly develop resistance to these agents, which are now widespread in clinically relevant bacteria due to spreading via horizontal gene transfer [5]. Continued successful application of sulfonamide antimicrobials involves specific approaches to overcome this limitation. One of them includes the application of sulfonamides as a part of combination therapy. In them, a number of drugs are used simultaneously leading to different ways of the folate metabolism disorders [5]. Another possibility is the introduction of sulfanilamide fragments into the macrocyclic structure, which leads to the localization of pharmaceutical groups and their solubility in physiological environments [6]. It is possible to reach high local concentration gradients of the drug by its application directly in the area of action, especially in case of external application.

Macrocycles, e.g., pillar[n]arenes and calix[n]arenes, are considered as promising platform for the design of novel drug agents due to their wide functionalization possibilities and ability to include pharmacophores as substituents in modification of the macrocyclic core or as participants of the host–guest complex [7,8,9]. Macrocycle molecule can carry various functional groups that bring hydrophilic/hydrophobic properties of the macrocycles and its activity in self-association yielding in formation of vesicles in solution or on the film interfaces [10,11,12]. Thus, the obtained supramolecular systems have the capability of responding to external stimuli such as photo, pH, and redox sensitivity. As a result, they demonstrate varying complex-forming properties [13]. The physicochemical properties of the macrocyclic compounds make them a promising platform for controlled drug delivery including antimicrobials [12,14,15]. Structural features of macrocycle carriers enhance stability improve biocompatibility and can promote therapeutic agent penetration into the action locus [16]. Such advantages were demonstrated by the synthesis of antibacterial drugs based on the pillar[n]arenes functionalized with an antimicrobial agent or involved in the complex with them [17]. In this work, sulfonamide-functionalized pillar[5]arene has been synthesized as a promising antibacterial drug and its antimicrobial and cytotoxic properties characterized. Application of macrocyclic platform in design of antimicrobial drug improved its bioavailability and drug targeting action.

## 2. Materials and Methods

Detailed information of equipment, methods, physical–chemical characterization is presented in electronic supporting information (ESI).

### 2.1. Synthesis of Compounds ***2**–**9***

Pillar[5]arene **1** was synthesized according to the literature’s procedure [18].

#### 2.1.1. 4,8,14,18,23,26,28,31,32,35-Deca-[thioethane(2′-(N,N-dimethyl)amino)ethoxy]-pillar[5]arene (**2**)

In the round-bottom flask equipped with a magnetic stirrer, 0.46 g (19.2 mmol) of sodium hydride was dissolved in 16 mL of anhydrous DMF, then 0.61 g (4.32 mmol) of 2,2-dimethylaminoethanethiol was added. After that, 0.4 g (0.24 mmol) of decabromoethoxypillar[5]arene **1** was added to the reaction mixture at 5–10 °C until fully dissolved. The reaction was carried out at 100 °C for 56 h in an argon atmosphere. The target product was isolated by precipitation from distilled water. The precipitate formed was centrifuged, and the aqueous solution was decanted. The precipitate was dissolved in 10 mL of methylene chloride, thereafter the target compound was concentrated using a rotary evaporator.

Yield: 0.29 g (62%), m.p. 154 °C

^1^H NMR (CDCl_3_): 1.75 s (20H, -CH_2_S-), 2.24 s (60H, -CH_3_), 2.52 t (20H, ^3^J_HH_ = 6.9 Hz, -SCH_2_-), 2.71 t (20H, ^3^J_HH_ = 6.9 Hz, -CH_2_N-), 3.77 s (10H, -CH_2_-), 3.93–4.20 m (20H, -OCH_2_-), 6.87 s (10H, ArH).

^13^C NMR (CDCl_3_): 29.45, 30.58, 32.16, 45.46, 59.50, 68.37, 115.61, 128.75, 149.77.

IR (v, sm^−1^): 3911 (-C_Ph_-H), 3775 (-C_Ph_-H), 3427 (C_Ph_-C_Ph_), 3003 (C-N), 2924 (C-N), 2865 (-CH_2_-, C_Ph_-O-CH_2_), 1657 (C_Ph_-C_Ph_), 1472 (-CH_2_-), 1201 (C_Ph_-O-CH_2_), 1095 (C_Ph_-O-CH_2_); 1026 (ring); 899 (-C_Ph_-H); 544 (-S-C-).

MS (MALDI-TOF): calc. [M^+^] *m*/*z* = 1921.2, Found [M^+^] *m*/*z* = 1922.2.

Found (%): C, 59.48; H, 8.29; N, 7.20; O, 8.45; S, 16.58. Calc. for C_85_H_160_N_10_O_10_S_10_. (%): C, 59.34; H, 8.39; N, 7.28; O, 8.32; S, 16.67.

#### 2.1.2. General Procedure for the Synthesis of Macrocycles **3**–**7**

In the round-bottom flask equipped with a magnetic stirrer, 0.15 g (0.078 mmol) of macrocycle **2** was dissolved in 7 mL of anhydrous DMF. Then 2.34 mmol (0.15 mL) of methyl iodide, or 0.19 mL of ethyl iodide, or 0.26 mL of ethyl 2-bromoacetate, or 0.64 g of 2-bromo-N-(4-sulfamoylphenyl)acetamide, or 0.41 mL of benzyl bromide was added dropwise to the reaction mixture until complete dissolution of the initial substances. The reaction was carried out at room temperature for 48 h in an argon atmosphere. Next, the products were isolated by precipitation from diethyl ether. The precipitate formed was separated by filtration on a filter funnel and washed with diethyl ether.

4,8,14,18,23,26,28,31,32,35-Deca-[thioethane(2′-(N,N,N-trimethyl)ammonium)ethoxy]-pillar[5]arene iodide (**3**).

Yield: 0.21 g (79%), m.p. 156 °C

^1^H NMR (DMSO-*d_6_*): 3.04 m (40H, -SCH_2_-, -CH_2_S-), 3.13–3.17 m (90H, -CH_3_), 3.65–3.75 m (30H, -NCH_2_-, -CH_2_-), 3.92–4.01, 4.22–4.37 m (20H, -OCH_2_-), 6.93 s (10H, ArH).

^13^C NMR (DMSO-*d_6_*): 26.92, 30.82, 31.41, 52.45, 64.48, 67.62, 114.90, 128.48, 149.03.

IR (v, sm^−1^): 3911 (-C_Ph_-H), 3775 (-C_Ph_-H), 3427 (C_Ph_-C_Ph_), 3003 (C-N^+^), 2924 (C-N^+^), 2865 (-CH_2_-, C_Ph_-O-CH_2_), 1657 (C_Ph_-C_Ph_), 1472 (-CH_2_-), 1201 (C_Ph_-O-CH_2_), 1095 (C_Ph_-O-CH_2_); 1026 (ring); 899 (-C_Ph_-H); 544 (-S-C-).

MS (ESI): calc. [M-7I^−^]^3+^ *m*/*z* = 986.84, Found [M-7I^−^]^3+^ *m*/*z* = 987.17, [M-6I^−^]^4+^ *m*/*z* = 708.65; [M-5I^−^]^5+^ *m*/*z* = 541.34.

Found (%): C, 37.75; H, 5.80; I, 37.95; N, 4.15; O, 4.85; S, 9.50. Calc. for C_105_H_190_I_10_N_10_O_10_S_10_. (%): C, 37.73; H, 5.73; I, 37.97; N, 4.19; O, 4.79; S, 9.59.

4,8,14,18,23,26,28,31,32,35-Deca-[thioethane(2′-(N,N-dimethyl, N-ethyl)ammonium)ethoxy]-pillar[5]arene iodide (**4**).

Yield: 0.20 g (75%), m.p. 158 °C

^1^H NMR (DMSO-*d*_6_): 1.26 t (30H, ^3^J_HH_ = 7.0 Hz, -CH_2_CH_3_), 3.05–3.18 m (100H, -SCH_2_-, -CH_2_S-, -CH_3_), 3.46 m (20H, -CH_2_S-), 3.64 q (20H, ^3^J_HH_ = 7.0 Hz, -CH_2_CH_3_), 3.64 t (20H, -CH_2_N-), 3.72 s (10H, -CH_2_-), 3.97, 4.27 m (20H, -OCH_2_-), 6.94 s (10H, ArH).

^13^C NMR (DMSO-*d*_6_): 8.12, 23.84, 30.78, 31.39, 49.50, 58.89, 62.14, 67.65, 114.93, 128.51, 149.02

IR (v, sm^−1^): 3438 (-C_Ph_-H), 2927 (C-N^+^), 2868 (-CH_2_-, C_Ph_-O-CH_2_), 1653 (C_Ph_-C_Ph_), 1462 (-CH_2_-), 1206 (C_Ph_-O-CH_2_), 1097 (C_Ph_-O-CH_2_); 1020 (ring); 917 (-C_Ph_-H).

MS (ESI): calc. [M^3+^] *m*/*z* = 1033.56, [M^4+^] *m*/*z* = 743.69, [M^5+^] *m*/*z* = 569.57, [M^6+^] *m*/*z* = 453.49, Found [M^3+^] *m*/*z* = 1033.50, [M^4+^] *m*/*z* = 743.61, [M^5+^] *m*/*z* = 569.50, [M^6+^] *m*/*z* = 453.40.

Found (%): C, 39.76; H, 6.13; I, 36.40; N, 3.99; O, 4.63; S, 9.09. Calc. for C_115_H_210_I_10_N_10_O_10_S_10_. (%): C, 39.66; H, 6.08; I, 36.44; N, 4.02; O, 4.59; S, 9.12.

4,8,14,18,23,26,28,31,32,35-Deca-[thioethane (2-(N,N-dimethyl, N-acetylethoxy)ammonium) ethoxy]-pillar[5]arene bromide (**5**).

Yield: 0.22 g (80%), m.p. 162 °C

^1^H NMR (DMSO-*d*_6_): 1.20 t (30H, ^3^J_HH_ = 7.1 Hz, -OCH_2_CH_3_), 3.07–3.20 m (40H, -SCH_2_-, -CH_2_S-), 3.32–3.35 s (60H, -N^+^CH_3_), 3.38–3.40 m (20H, -CH_2_N-), 3.70 s (10H, -CH_2_-), 3.93 t (20H, ^3^J_HH_ = 7.9 Hz, -OCH_2_-), 4.19 q (20H, ^3^J_HH_ = 7.03 Hz, -OCH_2_CH_3_), 4.61 s (20H, -CH_2_C=O-), 6.92 s (10H, ArH).

^13^C NMR (DMSO-*d*_6_): 13.86, 23.89, 30.84, 31.35, 51.07, 60.72, 62.18, 63.67, 67.50, 114.69, 128.38, 149.00, 164.90.

IR (v, sm^−1^): 2926 (-C_Ph_-H), 2870 (-CH_2_-, C_Ph_-O-CH_2_), 2760 (C-N^+^), 1739 (C=O), 1658 (C_Ph_-C_Ph_), 1496 (C_Ph_-C_Ph_), 1462 (-CH_2_-), 1201 (C_Ph_-O-CH_2_), 1094 (C_Ph_-O-CH_2_); 1021 (ring); 887 (-C_Ph_-H), 598 (-S-C-).

MS (ESI): calc. [M − 9Br^−^ + Na^+^]^10+^ *m*/*z* = 289.2, [M − 2Br^−^ + 16K^+^ + 8Na^+^]^26+^ *m*/*z* = 131.50, [M − 10Br^−^ + 2Na^+^]^12+^ *m*/*z* = 234.24, Found [M − 9Br^−^ + Na^+^]^10+^ *m*/*z* = 289.16, [M − 2Br^−^ + 16K^+^ + 8Na^+^]^26+^ *m*/*z* = 132.10, [M − 10Br^−^ + 2Na^+^]^12+^ *m*/*z* = 236.13.

Found (%): C, 45.14; H, 6.43; Br, 22.25; N, 3.89; O, 13.51; S, 8.78. Calc. for C_135_H_230_Br_10_N_10_O_30_S_10_. (%): C, 45.15; H, 6.45; Br, 22.24; N, 3.90; O, 13.36; S, 8.92.

4,8,14,18,23,26,28,31,32,35-Deca-[thioethane(2′-(N,N-dimethyl, N-methylcarbamate(N-4’-benzylsulfamide))ammonium)ethoxy]-pillar[5]arene bromide (**6**).

Yield: 0.23 g (63%), m.p. 159 °C

^1^H NMR (DMSO-*d*_6_): 3.13 m (40H, -CH_2_S-, -SCH_2_-), 3.24–3.39 m (60H, -CH_3_), 3.96 m (20H, -CH_2_N-), 4.25 s (10H, -CH_2_-), 4.51 m (20H, -OCH_2_-), 6.93 s (10H, ArH), 7.35 s (20H, -CH_2_C=O), 7.71–7.85 m (40H, PhH_Sulfanilamide_), 11.04 s (10H, -NH-).

^13^C NMR (DMSO-*d*_6_): 24.17, 28.61, 31.33, 51.31, 62.41, 64.19, 67.12, 114.30, 119.43, 126.86, 128.17, 139.55, 140.56, 149.10, 164.93.

IR (v, sm^−1^): 3173 (N–H amide), 3101 (N–H_amide_), 3029 (-C_Ph_-H), 2870 (-CH_2_-, C_Ph_-O-CH_2_), 1739 (C=O), 1692 (C_Ph_-C_Ph_), 1654 (C-N), 1592 (C_Ph_-C_Ph_), 1462 (-CH_2_-), 1201 (C_Ph_-O-CH_2_), 1152 (-C_Ph_-H); 1152 (S=O), 1095 (C_Ph_-O-CH_2_); 1021 (ring); 598 (-S-C-).

MS (ESI): calc. [M − 4Br^−^]^6+^ *m*/*z* = 736.47; [M − 10Br^−^ + 3K^+^ + H^+^ + Cl^−^]^13+^ *m*/*z* =323.47; [M − 10Br^−^ + 10K^+^ + 2H^+^ + 11Na^+^]^33+^
*m*/*z* = 142.33, Found [M − 4Br^−^]^6+^ *m*/*z* = 734.57, [M − 10Br^−^ + 3K^+^ + H^+^ + Cl^−^]^13^ *m*/*z* = 324.36, [M − 10Br^−^ + 10K^+^ + 2H^+^ + 11Na^+^]^33+^ *m*/*z* = 142.15.

Found (%): C, 43.27; H, 5.13; Br, 16.68; N, 8.56; O, 13.18; S, 13.18. Calc. for C_175_H_250_Br_10_N_30_O_40_S_10_. (%): C, 43.30; H, 5.19; Br, 16.46; N, 8.66; O, 13.18; S, 13.21.

4,8,14,18,23,26,28,31,32,35-Deca-[thioethane (2’-(N,N-dimethyl, N-benzyl) ammonium)ethoxy]-pillar[5]arene bromide (**7**).

Yield: 0.20 g (74%), m.p. 160.5 °C

^1^H NMR (DMSO-*d*_6_): 3.08 s (60H, -CH_3_), 3.19–3.29 m (40H, -CH_2_SCH_2_-), 3.67–3.86 m (30H, -CH_2_-, -CH_2_N-), 3.98, 4.30 m (20H, -OCH_2_-), 4.79 s (20H, -CH_2_Ph), 6.97 s (10H, ArH), 7.50 t (30H, ^3^J_HH_ = 7.1 Hz, Ph), 7.62 d (20H, ^3^J_HH_ = 6.6 Hz, Ph).

^13^C NMR (DMSO-*d*_6_): 28.57, 31.43, 34.36, 48.95, 63.04, 65.93, 67.51, 114.87, 128.07, 128.40, 128.92, 130.35, 133.01, 149.02.

IR (v, sm^−1^): 3399 (C_Ph_-C_Ph_), 2956 (C-N^+^), 2766 (-CH_2_-, C_Ph_-O-CH_2_), 2051 (C_Bz_ -H), 1656 (C_Ph_-C_Ph_), 1460 (-CH_2_-), 1205 (C_Ph_-O-CH_2_), 1096 (C_Ph_-O-CH_2_); 1024 (ring); 885 (-C_Ph_-H); 540 (-S-C-).

MS (ESI): calc. [M^2+^ − 2Br^−^ + K^+^ + Na^+^]^4+^ *m*/*z* = 882.35, Found [M^2+^ − 2Br^−^ + K^+^ + Na^+^]^4+^ *m*/*z* = 882.00.

Found (%): C, 54.83; H, 6.45; Br, 21.88; N, 3.78; O, 4.47; S, 8.59. Calc. for C_165_H_230_Br_10_N_10_O_10_S_10_. (%): C, 54.55; H, 6.38; Br, 21.99; N, 3.86; O, 4.40; S, 8.72.

#### 2.1.3. 4,8,14,18,23,26,28,31,32,35-Deca-[thioethane (2′-(N,N-dimethyl, N-propane-1-sulfonate)ammonium)ethoxy)-pillar[5]arene (**8**)

In the round-bottom flask equipped with a magnetic stirrer, 0.22 g (0.114 mmol) of macrocycle **2** was dissolved in 10 mL of anhydrous DMF. Then 0.28 g (2.28 mmol) of 1,3-propane sultone was added to the reaction mixture until complete dissolution of the initial substances. The reaction was carried out at room temperature for 48 h in an argon atmosphere. Next, the reaction mixture was precipitated in diethyl ether. After the following decantation, the sediment was dissolved in methanol and concentrated using the rotary evaporator.

Yield: 0.22 g (60%), m.p. 161 °C

^1^H NMR (D_2_O): 2.13 s (20H, -CH_2_CH_2_CH_2_SO_3_^−^), 2.88–2.90 m (20H, -CH_2_CH_2_CH_2_SO_3_^−^), 2.97–3.15 m (100H; -CH_3,_ -CH_2_S-, -SCH_2_-), 3.32–3.58 m (40H, -CH_2_N^+^CH_2_-), 3.85 s (10H, -CH_2_-), 4.15 m (20H, -OCH_2_-), 6.98 s (10H, ArH).

^13^C NMR (DMSO-*d*_6_): 28.88, 31.39, 34.36, 48.93, 51.00, 57.33, 63.13, 66.00, 67.50, 114.80, 128.49, 149.24.

IR (v, sm^−1^): 3434 (C-N^+^), 2931 (-C_Ph_-H), 2867 (-CH_2_-, C_Ph_-O-CH_2_), 1656 (C-N^+^), 1495 (C_Ph_-O-CH_2_); 1469 (C_Ph_-C_Ph_), 1436 (C_Ph_-C_Ph_), 1404 (-CH_2_-), 1387 (-CH_2_-), 521 (-S-C-), 1183 (-SO_2_-O^−^), 1031 (ring); 902 (-C_Ph_-H), 521 (-S-C-).

MS (MALDI-TOF): calc. [M + 6K^+^ + 4Na^+^]^10+^ *m*/*z* = 346.67, [M + 8Na^+^ + 4Li^+^]^12+^ *m*/*z* = 279.48, Found [M + 6K^+^ + 4Na^+^]^10+^ *m*/*z* = 346.2682, [M + 8Na^+^ + 4Li^+^]^12+^ *m*/*z* = 346.2682 [M + 6K^+^ + 4Na^+^]^10+^ *m*/*z* = 279.0934.

Found (%): C, 47.74; H, 6.99; N, 4.48; O, 20.43; S, 20.36. Calc. for C_125_H_220_N_10_O_40_S_20_. (%): C, 47.75; H, 7.05; N, 4.45; O, 20.35; S, 20.39.

#### 2.1.4. 4,8,14,18,23,26,28,31,32,35-Deca-[thioethane (2′-(N,N-dimethyl, N-acetate)ammonium)ethoxy]-pillar[5]arene (**9**)

In the round-bottom flask equipped with a magnetic stirrer, 0.3 g (0.107 mmol) of the macrocycle **5** was dissolved in 10 mL of anhydrous tetrahydrofuran (THF). Then, 0.8 g (2.14 mmol) of lithium hydroxide was dissolved in 2 mL of distilled water and added to the reaction mixture. The reaction was carried out at room temperature for 24 h. Next, the precipitate formed during the reaction was filtrated, and the filtrate was concentrated using the rotary evaporator.

Yield: 0.14 g (65%), m.p. 163 °C

^1^H NMR (DMSO-*d_6_*): 2.92–3.09 m (40H; -CH_2_S-, -SCH_2_-), 3.21 s (60H, -CH_3_), 3.64–3.75 m (30H; -CH_2_N-, CH_2_-), 3.81–3.98 m (30H; -OCH_2_-, -CH_2_COO^−^), 4.29 m (10H, -OCH_2_-), 6.94 s (10H, ArH).

^13^C NMR (DMSO-*d_6_*): 28.66, 30.78, 31.39, 49.50, 62.14, 67.65, 73.36, 114.93, 128.51, 149.02, 174.53.

IR (v, sm^−1^): 3563 (COO–H), 3184 (-C_Ph_-H), 2820 (-CH_2_-, C_Ph_-O-CH_2_), 2338 (C-N+), 1571 (C=O), 1496 (-CH_2_-), 1403 (C_Ph_-O-CH_2_), 1210 (C_Ph_-C_Ph_), 617 (-S-C-), 982 (ring); 838 (-C_Ph_-H).

MS (ESI): calc. [M + 6K^+^ + 4Na^+^]^10+^ *m*/*z* = 279.48, Found [M + 6K^+^ + 4Na^+^]^10+^ *m*/*z* = 279.0940.

Found (%): C, 55.21; H, 7.20; N, 5.65; O, 19.25; S, 12.69. Calc. for C_115_H_180_N_10_O_30_S_10_. (%): C, 55.18; H, 7.25; N, 5.60; O, 19.17; S, 12.81.

### 2.2. Biological Investigations

#### 2.2.1. Bacterial Strains and Cell Cultures

*Salmonella typhimurium* TA98 and TA100 laboratory strains as well as clinical isolates of: *Klebsiella pneumoniae*, *Staphylococcus epidermidis*, *Staphylococcus aureus*, *Pseudomonas aeruginosa* obtained from the Kazan Institute of Epidemiology and Microbiology (Kazan, Russia) were grown in LB-broth (tryptone—10 g/L; yeast extract—5 g/L; NaCl—5 g/L; pH 7.5). The day before the experiments the cultures were inoculated into a fresh medium to maintain the exponential growth phase. Biofilm formation was determined using Basal medium (BM), which is a modified SMM medium [19] with the addition of peptone up to 7 g/L.

Human lung adenocarcinoma A549 and LEK cells (both lines from Russian Collection of Cell Cultures of Vertebrates (CCCV)) were cultured in DMEM (Gibco, Waltham, MA, USA) supplemented with 10% FBS (BioSera, Nuaille, France), 2 mM L-glutamine, 100 mg/L penicillin and 100 mg/L streptomycin.

#### 2.2.2. MICs Determination

The MIC of the macrocycles and sulfanilamide was determined via a resazurin test in 96-well microtiter plates (SPL Life Sciences Co. Ltd., Gyeonggi-do, Korea) according to [20]. Briefly, the agents were subsequently diluted in L-broth (1:1). The concentration ranges were 1.2–300 µM for pillar[5]arenes and 19–4800 µM for sulfanilamide. Then the bacterial suspensions (1 × 10^8^ cells/mL final concentration) were added to wells and plates were incubated for 24 h at 37 °C in humidified atmosphere. After incubation the resazurin solution (10 µL, 1%) was added to the wells and plates incubated for several hours at 37 °C. A change in color from purple to pink or discoloration was considered as evidence of the microorganisms’ development in the well.

#### 2.2.3. BPC Analysis

To determine the biofilm prevention concentrations (BPC) of the macrocycles, the modified crystal–violet staining method was applied [21]. Briefly, the bacterial suspensions (1 × 10^6^ cells/mL) in BM-broth were seeded into 24-well culture plates (SPL Life Sciences Co., Ltd., Gyeonggi-do, Korea). Macrocycles were added to wells in the final concentrations range of 5–70 µM, and the *S. aureus* and *P. aeruginosa* plates were incubated under static conditions for 24 h at 37 °C. Then the liquid culture was removed from wells and plates were washed twice with PBS (pH 7.4) and dried. Then 1 mL of the 0.1% crystal–violet solution (Sigma Aldrich, Burlington, MA, USA) in 96% ethanol was added per well, followed by 20 min incubation. Next, the crystal violet solution was removed and the plates were washed 3 times with PBS. After at least 30 min air drying, 1 mL of 96% ethanol was added to resolubilize the crystal violet bound with biofilms, and the absorbance was measured at 570 nm with the BIO-Rad xMark Microplate spectrophotometer. Cell-free wells incubated with pure medium subjected to all staining manipulations were used as control. The concentration of macrocycle at which no biofilm development was detected was considered as BPC.

#### 2.2.4. Mutagenicity Assay

The mutagenicity of macrocycles was assessed using the Ames test with *Salmonella typhimurium* TA98 and TA100 strains as described [22]. The tested compound was considered to be mutagenic if the number of induced revertant colonies in the experiment was more than two-times higher than spontaneous [23].

#### 2.2.5. Cytotoxicity

MTT assay [24] was used to determine the changes in the viability of A549 and LEK cells under the action of macrocyclic compounds. Briefly, cells were seeded in 96-well plates with the density of 1 × 10^4^ cells per well and left overnight for the attachment at 37 °C and 5% CO_2_ in humidified atmosphere. Then the medium in wells was replaced with a fresh one with various pillar[5]arenes concentrations and plates were incubated for 48 h. After incubation, the medium in the wells was changed to an MTT (0.5 mg/mL final concentration) containing medium for 3 h. Then the medium was aspirated and 100 µL of dimethyl sulfoxide per well was added to dissolve the formazan crystals. The optical density of the formazan solution in the wells was measured using BioRadxMarkTM Microplate Spectrophotometer (Bio-Rad Laboratories, Hercules, CA, USA) at a wavelength of 570 nm.

#### 2.2.6. Statistical Analysis

All experiments were performed in biological triplicates (i.e., newly prepared cultures and medium) with at least three repeats in each run. For all tests significant differences were reported at *p*  <  0.05 using the nonparametric Mann–Whitney U-test.

## 3. Results and Discussion

### 3.1. Synthesis and the Aggregation Characteristics of Water-Soluble Ammonium Derivatives of pillar[5]arene

Introduction of pharmacophore fragments into the structure of pillar[5]arene can significantly alter its biological properties. Presence of macrocyclic cavity and spatially pre-organized structure promotes the formation of inclusion complexes with various drugs, shows reduced toxicity, and prolongs their action period of the drug [25]. In this study, we propose the preparation of a series of water-soluble decasubstituted pillar[5]arene derivatives containing cationic pharmacophore (quaternary ammonium groups) and sulfonamide (streptocide) as substituents.

Discovered in the 1930s, quaternary ammonium salts (QAS) are up to now the most common antimicrobial and disinfectant agents [26]. Meanwhile, modern research confirms their antifungal, antiviral and antibiofilm properties [27,28]. The Menshutkin reaction is mostly used for the QAS synthesis. In this work, we started from the synthesis of decabromoethoxypillar[5]arene **1** according to the literature’s method [18]. Then, macrocycle **1** reacted with commercially available 2,2-dimethylaminoethanethiol in the presence of sodium hydride. The synthesis was carried out in anhydrous dimethylformamide (DMF) at 90 °C. Macrocycle **2** was isolated in the yield of 62%. A series of target ammonium derivatives of pillar[5]arene were synthesized on the basis of macrocycle **2**. Macrocycles **3**–**5** were obtained with the yields of 75–80% via Menshutkin reaction in the interaction of macrocycle **2** with iodomethane, ethyl iodide and ethyl-2-bromoacetate in anhydrous DMF at room temperature. The interaction of 2-bromo-N-(4-sulfamoylphenyl)acetamide with macrocycle **2** in anhydrous DMF at room temperature resulted in formation of target macrocycle **6** with 63% yield. The alkylating agent, 2-bromo-N-(4-sulfamoylphenyl)acetamide, was obtained according to a literature method [29]. Further, compound **2** was converted to macrocycle **7** (Figure 1) by the reaction with benzyl bromide**.** Betaine derivatives **9** and **8** were synthesized by subsequent hydrolysis of macrocycle **5** and the reaction of pillararene **2** with 1,3-propane sultone, correspondingly. The structure of all obtained decasubstituted pillar[5]arenes 2–**9** was confirmed by a number of physical methods (NMR ^1^H, NMR ^13^C{^1^H}, IR-, MALDI- and ESI-mass spectroscopy). Appropriate data are presented in ESI.

For organic compounds with potential biological activity, colloidal stability can have a significant impact on their efficacy [30]. The discovery of new drugs has been hampered for a very long time by colloidal aggregates formed by many small organic molecules in aqueous solution [31]. These colloids, ranging in size from 50 to over 800 nm, are formed spontaneously and reversibly in an aqueous buffer when subjected to a critical concentration of an aggregation similar to the critical concentration of micelle formation [32]. These colloids, formed as a result of self-assembly, are capable of adsorbing onto the surface of membrane proteins, causing partial denaturation and resulting in non-specific inhibition of enzymes and other proteins [33]. It is widely acknowledged that the random inhibition resulting from the aggregation of small molecules is the fundamental cause of false positives in a high-throughput as well as virtual screening [32]. Without precipitating or forming colloidal systems or aggregates, drugs can hold their efficacy for a long time [34]. Therefore, the aggregation ability of synthesized compounds **3**–**6, 8, 9** in water was investigated by dynamic light scattering (DLS) in the concentration range of 10–1000 µM (ESI, Appendix A). Macrocycle **7** containing lipophilic benzyl moieties was poorly soluble in water. This complicates its use as an antibacterial agent. Further, DLS method showed for aqueous solution of pillar[5]arenes **4**, **6**, **8** no formation of stable self-associates in the whole range of concentrations investigated. Polydispersity index (PDI) values for **4**, **6**, **8** ranged from 0.38 to 0.62, while the ζ—potential of the system did not exceed 4 mV (Table 1, ESI, Appendix A), indicating the absence of stable colloidal systems. However, macrocycles **3**, **5**, **9** form self-associates in the range of 10–1000 µM concentrations in water. Thus, the PDI for **3**, **5**, **9** ranged from 0.27 to 0.37 (10 µM) and the ζ- potential systems were in the range of 18–47 mV (Table 1, ESI, Appendix A). Such high ζ-potential values indicate the formation of a stable colloidal system. It is evident that the aggregation properties of macrocycles in this series of pillar[5]arenes (**3**–**6**, **8** and **9**) are influenced by the nature of the substituent in the ammonium nitrogen atom and the nature of the anion. Thus, analysis of experimental data showed that compounds **4**, **6**, **8** are present in the solution individually.

### 3.2. Biologic Activity

#### 3.2.1. Antibacterial Activity

At the next step of the study, the minimum inhibitory concentrations (MICs) of the macrocycle **2** and of water-soluble derivatives **3**–**6, 8, 9** were determined against *Salmonella typhimurium* TA 98, and clinical isolates of *Klebsiella pneumoniae*, *Staphylococcus epidermidis*, *Staphylococcus aureus*, *Pseudomonas aeruginosa*. Sulfonamide was used as a standard (Table 2). Due to solubility limitations of some macrocycles observed under physiologic conditions, the maximum concentrations of pillar[5]arenes tested were limited by 300 µM (581.10–1459.92 µg/mL depending on compound), and that of sulfonamide by 4800 µM (826.56 µg/mL). Parent macrocycle **2** as well as macrocycles **3**–**5**, **8**, **9** tending to aggregation did not inhibit microbial growth (MIC ≥ 300 µM) in the whole range of concentrations studied. Macrocycle **9** inhibited the growth of *S. typhimurium* TA98, *K. pneumoniae*, *S. epidermidis*, *P. aeruginosa* at its concentration of 300 µM (754.62 µg/mL). It is worth noting that the bacteria showed high-resistance to sulfonamide as could be expected. It inhibited their growth only at maximum concentration of 4800 µM among those tested. *P. aeruginosa* (MIC = 2400 µM; 413.28 µg/mL) was the only exception. Meanwhile, pillar[5]arene **6** functionalized with sulfonamide fragments exerted a significant antibacterial effect and inhibited growth of all tested microorganisms at concentrations of 37.5–75 µM (182.49–364.98 µg/mL) except that of *K. pneumonia* (MIC = 150 µM; 729.96 µg/mL).

The action of functionalized macrocycle **6** cannot be attributed only to the separation of sulfonamide from the macrocycle core by probable biodegradation because its amount in the complex agent in the wells was below the MIC of the pure drug. Hence, we hypothesize that the inclusion of sulfonamide into the macrocycle improves its bioavailability to microorganisms. The sensitivity of bacteria to the sulfonamide as part of the macrocycle increases by approximately an order of magnitude. It is important that this activity was found against clinical isolates of microorganisms, including *P. aeruginosa* and *K. pneumonia* which have a high natural resistance to agents that disrupt folic acid metabolism [5].

#### 3.2.2. Antibiofilm Activity

The ability of clinically important microorganisms to form biofilms is an essential factor in their resistance to antimicrobial agents and environmental factors. Next stage of our research involved determination of the ability of macrocycle **6** to inhibit the formation of microbial biofilms. Clinical isolates of *S. aureus* and *P. aeruginosa* were selected as model samples. These species are typical causative pathogens of nosocomial infections [35]. The results of evaluating the antibiofilm effect of macrocycle **6** functionalized with sulfonamide fragments are presented in Table 3.

Macrocycle **6** inhibits the biofilm formation of *S. aureus* and *P. aeruginosa*. Biofilm prevention concentrations (BPCs) were equal to 20 µM (97.33 µg/mL) and 50 µM (243.32 µg/mL), respectively. Since MIC and BPC did not differ significantly, it can be concluded that compound **6** has no specific antibiofilm activity. Its influence on the biofilm formation can be due to its toxicity against test microorganisms [36].

#### 3.2.3. Cytotoxicity and Mutagenicity

The observed antibacterial activity of macrocycle **6** makes it possible to consider it as a promising therapeutic agent. For this reason, it is necessary to evaluate its biocompatibility properties. We have characterized in vitro mutagenicity of compounds **2** and **6** in the Ames test and their cytotoxicity in the MTT test.

No mutagenicity of initial macrocycle **2** and of compound **6** was detected in the Ames test without metabolic activation using *Salmonella typhimurium* TA98 and TA100 strains. The concentrations of the macrocycles were up to 300 µM (581.10 µg/mL) for macrocycle **2** and 300 µM (145.99 µg/mL) for compound **6** (Table 4). The concentration of compound **6** 300 µM (1459.92 µg/mL) exceeding the MIC was excluded from the study.

The MTT test revealed that pillar[5]arene **2** did not have the ability to affect the viability of A549 human lung adenocarcinoma cells and bovine embryonic lung epithelial cells (LEKs) in the range of concentrations tested (3–300 µM) (Figure 1). Macrocycle **6** exhibited minor cytotoxicity only at its highest concentration (300 µM, or 1459.92 µg/mL). It reduced viability of the A549 and LEK cells by 18.42 ± 6.78% and 14.88 ± 5.11%, respectively.

Thus, we can preliminarily conclude on the potential suitability of macrocycle **6** for direct application in systemic therapeutic of humans and animals.

## 4. Conclusions

In this study, a novel biocompatible macrocyclic structure **6** based on a water-soluble pillar[5]arene derivative containing ammonium and streptocide fragments was designed and characterized. It showed a higher inhibitory effect towards Gram-positive and Gram-negative microorganisms against that of sulfonamide. The MICs of compound **6** for *Salmonella typhimurium* TA 98, *Klebsiella pneumoniae*, *Staphylococcus epidermidis*, *Staphylococcus aureus*, *Pseudomonas aeruginosa* were in the range of 75–150 µM. Pillar[5]arene **6** also showed the ability to inhibit the biofilm formation of *S. aureus* and *P. aeruginosa*, the BPCs were equal 20 µM and 50 µM, respectively. Meanwhile, the results of the MTT assay showed that macrocycle **6** exhibited slight cytotoxicity only at its highest concentration tested (300 µM). The obtained results, taking into account the absence of mutagenic activity of compound **6**, open a wide range of opportunities for using drug-modified macrocyclic compounds as new generation antibacterial and antibiofilm agents.

## Data Availability

Data is contained within the article or Appendix A.

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
