# Peer review of "The Synthesis and Antibacterial Properties of Pillar[5]arene with Streptocide Fragments"

_pharmaceutics, 2023, doi:10.3390/pharmaceutics15122660_

Round 1

Reviewer 1 Report

Comments and Suggestions for Authors

The Authors report on synthesis of several pillar[5]arene derivatives and their primary investigation in microbial and cytotoxicity assay. In general I find this study too preliminary and superficial to be published in Pharmaceutics. There are only several compounds synthesized by relatively straightforward methods and tested in primary assays. In this context, this study is notl likely to have impact in the field. I also have other comments for the Authors that are listed below:

1) The abstract states “The properties of classic antimicrobials such as sulfonamides, already of little use". This is a too strong sentence. Sulfonamides still do have use in treatment of infections.

2) Introduction: It would appear that according to the Authors the way to combat the resistance of bacteria against sulfonamide is “to increase the bioavailability of sulfonamides, limited by their low solubility in physiological media” which is not clear. Usually, the resistance is referred to a characteristics of a microorganism which is developed upon exposure to a harmful agents and is not a matter of ADME/PK issues of the compound.

3) The title is misleading. One would expect that the antibacterial sulfonamides are to be improved. Here, only one sulfonamide derivative 6 is presented, without any proof or speculation that the mechanism of action of the synthesized macromolecule matches this of the original sulfonamide. 

Supplementary information:

1) "The concentration of sample solutions was 3–5%.": It is unusual to report NMR concentrations in %, but rather in mM

2) The spectra should be provided in full ranges to allow for purity evaluation. If this causes peaks to be barely visible, magnification of the spectra should be provided, displaying regions where the peaks appear. Integrations for all signals should also be given (eg. what is ~8.0 ppm in figure S1?).

3) Some compounds do not appear to be pure.  I suspect that there is DMF contaminations in the samples, as seen by 1H NMR. Which is not acceptable. More, DMF is not a biologically inert rsolvent. Impurities are also visible in carbon spectra.

Comments on the Quality of English Language

The usage of English is accteptable, however the manuscript could use some polishing.

Author Response

Referee: 1

The Authors report on synthesis of several pillar[5]arene derivatives and their primary investigation in microbial and cytotoxicity assay. In general I find this study too preliminary and superficial to be published in Pharmaceutics. There are only several compounds synthesized by relatively straightforward methods and tested in primary assays. In this context, this study is not likely to have impact in the field. I also have other comments for the Authors that are listed below:

1) The abstract states “The properties of classic antimicrobials such as sulfonamides, already of little use". This is a too strong sentence. Sulfonamides still do have use in treatment of infections.

Answer: Thank you for your comment! This sentence has been used to highlight the serious limitations in the use of this class of drugs due to the increasing resistance of microorganisms’ target strains. Extensive therapeutic experience with sulfonamides shows that they have significant side effects associated with damage to organs and organ systems, including the cardiovascular, urinary and nervous systems. Among the listed side effects, the most dangerous are urinary complications (stones, sand in the kidneys and ureters) and decreased immune function of the body due to leucopenia. Stones and sand in the kidneys and ureters can lead to anuria, while a decrease of white blood cells and granulocytes can reduce the body's resistance to new infections [Supuran, C. T. (2017). sulfonamides. Molecules, 22(10), 1642.]. Indeed, potentiated sulfanilamides are still widely used [Treves-Brown, K.M. (2000). Potentiated Sulfonamides. In: Applied Fish Pharmacology. Aquaculture, vol 3. Springer, Dordrecht. https://doi.org/10.1007/978-94-017-0761-9_9]. Due to the gradual deterioration of antibiotic resistance in general, sulfonamides can be used as inexpensive antibacterial drugs, given the current knowledge of their side effects [Sköld, O. (2000). Sulfonamide resistance: mechanisms and trends. Drug resistance updates, 3(3), 155-160.].

To clarify that point, we have replaced the sentence:

“The properties of classic antimicrobials such as sulfonamides, already of little use, can be improved through the application of supramolecular chemistry.”  

with:

“Supramolecular chemistry tools can overcome the limited bacterial resistance and side effects of classical sulfonamides that hinder their use in therapy.”

2) Introduction: It would appear that according to the Authors the way to combat the resistance of bacteria against sulfonamide is “to increase the bioavailability of sulfonamides, limited by their low solubility in physiological media” which is not clear. Usually, the resistance is referred to a characteristics of a microorganism which is developed upon exposure to a harmful agents and is not a matter of ADME/PK issues of the compound.

Answer: An important feature of the microorganism's life cycle is the formation of colonies with further complication and development of biofilms that are tolerant to the action of antibiotics. The process of biofilm destruction differs from the action of the drug on planktonic bacteria. It is related not only to the characteristics of the microorganism, but also to the ability of the drug to penetrate and act within the biofilm, which corresponds to its bioavailability and solubility. Due to the low physiological solubility of sulfanilamide, pathogenic microorganisms are often resistant to the concentration of the drug reached in the body or its individual tissues.

An effective way to combat resistance is to selectively increase the concentration of the drug at the target site (via solubility and increased bioavailability) where the microorganism is unable to overcome its action. In this case, the term "stability" is not related to the specific characteristics of the microorganism, but reflects the resistance of the pathogen to a certain concentration of the drug. Thus, by increasing bioavailability, we can provide the microorganism with the necessary concentration of drug that it cannot overcome. Increasing the solubility, bioavailability and activity of a drug against microbial cells can be achieved by modifying it (for example, by incorporating it into a macrocyclic polymer platform) [Shurpik, D. N., Padnya, P. L., Stoikov, I. I., & Cragg, P. J. (2020). Antimicrobial activity of calixarenes and related macrocycles. Molecules, 25(21), 5145.].

In this work, we follow this principle by incorporating sulfanilamide fragments into the macrocyclic platform of the compound 2. All the obtained water-soluble macrocycles 3-9, with the exception of the streptocide derivative 6, do not show pronounced antimicrobial properties and act as reference compounds containing quaternary ammonium groups. The results confirm the need in sulfanilamide substitutes in the ammonium-derived pillar[5]aren for the expression of inhibitory properties for the studied strains of microorganisms. The antimicrobial behavior of macrocycle 6 can be explained by the action of macrocycle-free sulfanilamide fragments formed during biodegradation. This is consistent with the assumption that streptocide is more soluble when associated with the macrocyclic platform. However, the molecular basis of the observed effect is still unknown and requires complex research. 

In the manuscript the fragment:

“Another one is to increase the bioavailability of sulfonamides, limited by their low solubility in physiological media [6].”

has been replaced by:

“Another possibility is the introduction of sulfanilamide fragments into the macrocyclic structure, which leads to the localization of pharmaceutical groups and their solubility in physiological environments [6].”

3) The title is misleading. One would expect that the antibacterial sulfonamides are to be improved. Here, only one sulfonamide derivative 6 is presented, without any proof or speculation that the mechanism of action of the synthesized macromolecule matches this of the original sulfonamide. 

Answer: Our aim was to clarify and specify the name of the script. So, the name was changed to “The synthesis and antibacterial properties of pillar[5]arene with streptocide fragments.”

Supplementary information:

1) The concentration of sample solutions was 3–5%.: It is unusual to report NMR concentrations in %, but rather in mM

Answer: Required substitutions were produced in the supplementary information’s text: “The concentration of sample solutions was 5-10 mM.”

2) The spectra should be provided in full ranges to allow for purity evaluation. If this causes peaks to be barely visible, magnification of the spectra should be provided, displaying regions where the peaks appear. Integrations for all signals should also be given (e.g. what is ~8.0 ppm in figure S1?).

Answer: Thank you for the feedback. The compounds received additional cleaning, then the macrocycles’ spectra were re-registered and presented in the ESI file in extended format from 0-12 ppm. All signals have been integrated and correlated.

3) Some compounds do not appear to be pure. I suspect that there is DMF contaminations in the samples, as seen by 1H NMR. Which is not acceptable. More, DMF is not a biologically inert solvent. Impurities are also visible in carbon spectra.

Answer: DMF was eliminated from the compounds and macrocycles’ spectra were re-registered to present in an extended format in the ESI file from 0-12 ppm. In addition, biological experiments were carried out on purified compounds which were within the range of the initial values (MIC) of compounds with DMF impurities. This fact proves that low levels of DMF impurities do not (or only slightly) affect the biological activity of macrocyclic substrates.

Reviewer 2 Report

Comments and Suggestions for Authors

Comments on the Quality of English Language

Very poor English

Author Response

Referee: 3

This article presents some data for synthesis and biological activities of newly synthesized pillar[5]arenes. I have the following remarks to the authors:

1.English language of the paper is very poor. It needs to be completely corrected. There is no passive voice, but the most of the manuscript is written in the active voice

Answer: English was revised and grammatical constructions were taken into the careful consideration.

2.The title is extremely long.  Why the title needs of: Enhancing the activity of sulfonamides against opportunistic bacteria. In my opinion only second part has to be presented: The synthesis and antibacterial properties of pillar[5arene with streptocide fragments. This part describes perfectly the content of the work

Answer: The reviewer's note has been thoroughly examined and the title has been modified to reflect it.

Updated title: «The synthesis and antibacterial properties of pillar[5]arene with streptocide fragments».

3.Authors are prepared their article very carelessly: Why some number 1 presents in the first affiliation after Russia? There are multiple full stops after many sentences. Why in some long sentences there are multiple sentences separated by; for. ex. line 128. There are sentences in 4 lines (see lines 134-137) which are absolutely incomprehensible. Please use short and clear sentences.

Answer: The error was fixed. The article was revised and corrected.

4.Authors said in the Abstract section that sulfonamide are already of little use (line 14). So, if it is true why they introduce sulfonamide part in their conjugates (line 21)? Maybe they have to include the word classical sulfonamides in order to avoid this misunderstanding.

Answer: To clarify that, we replaced the sentence

“The properties of classic antimicrobials such as sulfonamides, already of little use, can be improved through the application of supramolecular chemistry.”

with:

“Supramolecular chemistry tools can be used to overcome the limited bacterial resistance and side effects associated with traditional sulfonamide therapies that hinder their clinical use.”

5.In line 22 authors said: and also had the ability to prevent biofilm formation of S. aureus and P. aeruginosa. Is this fact is revealed during current study or it is known from the literature. If it is currently found they have to put NOT had but is revealed. If it is known fact its place is not in the Abstract section but in the discussion section with appropriate citation.

Answer: Thank you for this attention, we have corrected the text.

In the manuscript, the fragment:

“It was shown that macrocycle functionalized with sulfonamide residues exhibited antimicrobial activity an order higher than pure streptocide and also had the ability to prevent biofilm formation of S. aureus and P. aeruginosa.”

was replaced by:

“It was shown that macrocycle functionalized with sulfonamide residues exhibited antimicrobial activity an order higher than pure streptocide and also revealed the ability to prevent biofilm formation of  S. aureus and P. aeruginosa.”

6.In the Keywords authors put antimicrobial and antibiofilm. These two words are adjectives and they need a substantive after, so please include activity, i.e. antimicrobial activity and antibiofilm activity

Answer: Thank you for your comment! We have made the necessary corrections in the keywords.

  1. In line 32: Sulfonamides can inhibit the development of Gram-positive…. REMOVE can, i.e. Sulfonamides inhibit the development of Gram-positive ……….

Answer: The correction has been made in the manuscript.

8.In line 31 what authors mean with systematic activity? What kind of activity is systematic?

Answer:  In the manuscript, the fragment:

“Sulfonamides-derived drugs are still used in medical and veterinary practice due to their systemic activity [2, 3]. Sulfonamides inhibit the development of Gram-positive and Gram-negative bacteria, as well as some protozoa [4].”

was replaced by:

“Sulfonamides-derived drugs are still used in medical and veterinary practice due to their systemic and local action [2, 3]. Sulfonamides have been shown to inhibit the growth of Gram-positive and Gram-negative bacteria, as well as some protozoa [4].”

9.In line 38 replace usage with application.

Answer: The correction has been made in the manuscript.

10.In line 45 platform to be replaced by matrix, because in 2 sentences the word is used several times.

Answer: The macrocyclic platform is crucial in terms of terminology. Substituting it with the word "matrix" leads to a loss of content meaning. However, we have taken into account the reviewer's feedback and eliminated some unnecessary repetition in the text.

“Macrocycles, e.g. pillar[n]arenes and calix[n]arenes, are considered as promising platform for design of novel drug agents due to their wide functionalization possibilities and ability to include pharmacophores as substituents in modification of the macrocyclic core or as participants of the host-guest [7-9]. Macrocycle molecule can carry various functional groups that bring hydrophilic/hydrophobic properties of the macrocycles and its activity in self-association yielding in formation of vesicles in solution or on the film interfaces [10-12].”

11.In line 48 give it to be replaced by bring

Answer: The correction has been made in the manuscript.

12.In line 51 what authors mean with different properties? Do they mean specific properties?

Answer: Yes, supramolecular ensembles based on soluble pillar[n]arenas demonstrate self-assembly into particles, vesicles, or films at the phase separation boundary under various system parameters and external stimuli such as photo-, pH-, and redox-sensitivity. Their interaction with the guest is also influenced by these stimuli. The term "different properties" refers to this context.

In the manuscript, the fragment:

“The obtained systems can depends on external stimuli and exhibit different properties under different conditions [13].”

was replaced by:

“Thus, the obtained supramolecular systems have the capability of responding to external stimuli such as photo, pH, and redox sensitivity. As a result, they demonstrate varying complex-forming properties [13].”

13.In line 87 is written that procedure is described below. Where is below? After 1 line, 2 lines or 55 lines? It is finally in the Material and Methods section

Answer:  The literature describes this synthesis procedure, and for clarity the following sentence

“The initial alkylating agent, 2-bromo-N-(4-sulfanylphenyl)acetamide, was previously obtained according to the procedure below [23].”

was replaced by:

“The alkylating agent, 2-bromo-N-(4-sulfanylphenyl)acetamide, was obtained according to a literature method [23].”

14.Scheme 1 does not present synthesis of macrocycles. It presents substitutions in the lateral chains of macrocycle

Answer:  Scheme of synthesis for compounds 2-9 in manuscript was expanded and replaced.

Scheme 1. Synthesis of the macrocycles 2-9.

15.In line 98 is written: aggregation stability can be an important factor for the efficacy of drugs. Why? Where it is proven? Some citation for this fact or some explications missed. In addition, after next sentence a literature [24] is cited which is absolutely not related with this statement.

Answer:  It is noteworthy that the American Chemical Society (ACS) has developed instructions for sending articles to its journals that describe the biological activity of compounds. These instructions stipulate that the presence of aggregators and unwanted fragments in such compounds shall be checked [Aldrich, C., Bertozzi, C., Georg, G. I., Kiessling, L., Lindsley, C., Liotta, D., ... & Wang, S. (2017). The ecstasy and agony of assay interference compounds. ACS Chemical Neuroscience, 8(3), 420-423.]. As experimental data have accumulated, molecules with potential biological activity that undergo spontaneous or reversible aggregation in water buffer have been classified as Pan-assay Interference Compounds (PAINS) [Baell, J. B., & Holloway, G. A. (2010). New substructure filters for removal of pan assay interference compounds (PAINS) from screening libraries and for their exclusion in bioassays. Journal of medicinal chemistry, 53(7), 2719-2740.]. The discovery of new drugs has been hampered for a very long time by colloidal aggregates formed by many small organic molecules in aqueous solution [Kaya, I., & Colmenarejo, G. (2020). Analysis of Nuisance Substructures and Aggregators in a Comprehensive Database of Food Chemical Compounds. Journal of Agricultural and Food Chemistry, 68(33), 8812-8824.]. These colloids, ranging in size from 50 to over 800 nm, form spontaneously and reversibly in an aqueous buffer when subjected to critical concentration of aggregation similar to the critical concentration of micelle formation [Irwin, J. J., Duan, D., Torosyan, H., Doak, A. K., Ziebart, K. T., Sterling, T., ... & Shoichet, B. K. (2015). An aggregation advisor for ligand discovery. Journal of medicinal chemistry, 58(17), 7076-7087.]. These colloids, formed as a result of self-assembly, are capable of adsorbing onto the surface of membrane proteins, causing partial denaturation and resulting in non-specific inhibition of enzymes and other proteins [Feng, B. Y., & Shoichet, B. K. (2006). A detergent-based assay for the detection of promiscuous inhibitors. Nature protocols, 1(2), 550-553.]. It is widely acknowledged that the random inhibition resulting from the aggregation of small molecules is the fundamental cause of false positives in high-throughput as well as virtual screening [Irwin, J. J., Duan, D., Torosyan, H., Doak, A. K., Ziebart, K. T., Sterling, T., ... & Shoichet, B. K. (2015). An aggregation advisor for ligand discovery. Journal of medicinal chemistry, 58(17), 7076-7087.].

In the manuscript the following sentence:

“In the case of antibacterials, aggregation stability can be an important factor for the efficacy of drugs and their ability to maintain the necessary concentrations in solution to achieve the required effect.”

was replaced by:

“For organic compounds with potential biological activity, colloidal stability can have a significant impact on their efficacy [24]. The discovery of new drugs has been hampered for a very long time by colloidal aggregates formed by many small organic molecules in aqueous solution [25]. These colloids, ranging in size from 50 to over 800 nm, form spontaneously and reversibly in an aqueous buffer when subjected to critical concentration of aggregation similar to the critical concentration of micelle formation [26]. These colloids, formed as a result of self-assembly, are capable of adsorbing onto the surface of membrane proteins, causing partial denaturation and resulting in non-specific inhibition of enzymes and other proteins [27]. It is widely acknowledged that the random inhibition resulting from the aggregation of small molecules is the fundamental cause of false positives in high-throughput as well as virtual screening [26].”

16.At many places the intervals of experimental concentrations are presented in M and further the exact values are presented in µg/mL. Please unify. In addition, in all descriptions of the experimental procedure mL is written as ml, please correct.

Answer: The reviewer's suggestions have been fully considered. Throughout the article and ESI-file, necessary changes from "ml" to "mL" have been made.

  1. In line 108 is written: This seems to be due to the higher lipophilicity of compounds 3, 5, 7 compared to 4, 6, 8. Is it seem or author prove it with their experiments?

Answer: In the manuscript the fragment:

“Further, it was shown using DLS that pillar[5]arenes 4, 6, 8 do not form stable self-associates in the whole range of concentrations investigated in water. However, macrocycles 3, 5, 9 form stable self-associates in the concentration of 10 µМ, ζ = 18-47 mV (Table 1). This seems to be due to the higher lipophilicity of compounds 3, 5, 9 compared to 4, 6, 8. The analysis of experimental data showed that compounds 4, 6, 8 are in individual form in solution.”

was replaced by:

“Further, DLS method showed for aqueous solution of pillar[5]arenes 4, 6, 8 no formation of stable self-associates in the whole range of concentrations investigated. Polydispersity index (PDI) values for 4, 6, 8 ranged from 0.38 to 0.62, while ζ - the potential of the system did not exceed 4 mV (Table 1, ESI, Table S1), indicating the absence of stable colloidal systems. However, macrocycles 3, 5, 9 form self-associates in the range of 10-1000 µM concentrations in water. Thus, the PDI for 3, 5, 9 ranged from 0.27 to 0.37 (10 µM) and the ζ-system potential was in the range of 17-47 mV (Table 1, ESI, Table S1). Such high ζ-potential values indicate the formation of a stable colloidal system. It is evident that the aggregation properties of macrocycles in this series of pillar[5]arenes (3-6, 8 and 9) are influenced by the nature of the substitute in the ammonium nitrogen atom and the nature of the anion. Thus, analysis of experimental data showed that compounds 4, 6, 8 are present in the solution individually.”

18.In line 118 for have to be replaced by against.

Answer: Correction has been made in the manuscript.

19.The presented experimental procedures are not well described and nobody cannot understand some places. For ex.:

-           Line 196 - The precipitate was dissolved in 10 ml of methylene chloride, evaporated on a rotary evaporator. Who is evaporated? In addition, in the same procedure DMF is already abbreviated and here it is again with full name why? Further, line 193-194: was added slowly at 5–10°C. The reaction was carried out at 100°C for 56 hours in an argon atmosphere. We are still at 5–10°C. When we start to heat to 100oC? Immediately after addition or 5 minutes later or after dissolving of compounds? At all, the description of all procedures is a mess.

-           Line 301: Next, the precipitate that formed was dissolved in water, and the filtrate was evaporated on a rotary evaporator. How we obtain the filtrate? There is not a filtration procedure mentioned in this text?

Answer: A number of changes were carried out to enhance the academic writing quality of the methods for the synthesizing compounds. Full solvent names were omitted, reaction temperature conditions were specified, and procedures for separating the final products were clearly delineated. The section "Materials & Methods" was revised, and the compound synthesis techniques were substituted.

“3.1.1. 4,8,14,18,23,26,28,31,32,35-deca-[thioethane(2'-(N,N-dimethyl)amino)ethoxy]- pillar[5]arene (2).

In the round bottom flask equipped with magnetic stirrer, 0.46 g (19.2 mmol) of sodium hydride were dissolved in 16 mL of anhydrous DMF, then 0.61 g (4.32 mmol) of 2,2-dimethylaminoethanethiol were added. After that, 0.4 g (0.24 mmol) of decabromethoxypillar[5]arene 1 was added to the reaction mixture at 5–10°C until full dissolving. The reaction was carried out at 100°C for 56 hours in an argon atmosphere. The target product was isolated by precipitation from distilled water. The precipitate formed was centrifuged, and the aqueous solution was decanted. The precipitate was dissolved in 10 mL of methylene chloride, thereafter the target compound was concentrated on a rotary evaporator.”

“3.1.2. General procedure for the synthesis of macrocycles 3-7

In the round bottom flask equipped with magnetic stirrer, 0.15 g (0.078 mmol) of macrocycle 2 were dissolved in 7 mL of anhydrous DMF. Then 2.34 mmol (0.15 ml) of methyl iodide or 0.19 mL of ethyl iodide or 0.26 mL of ethyl 2-bromoacetate or 0.64 g of 2-bromo-N-(4-sulfamoylphenyl)acetamide or 0.41 mL of benzyl bromide were added dropwise to the reaction mixture until complete dissolution of the initial substances. The reaction was carried out at room temperature for 48 hours an argon atmosphere. Next, the products were isolated by precipitation from diethyl ether. The precipitate formed was separated by filtration on a filter funnel and washed with diethyl ether.”

“3.1.3. 4,8,14,18,23,26,28,31,32,35- deca-[thioethane (2'-(N,N - dimethyl, N-propane-1-sulfonate))-pillar[5]arene (8).

In the round bottom flask equipped with magnetic stirrer, 0.22 g (0.114 mmol) of macrocycle 2 was dissolved in 10 ml of anhydrous DMF. Then 0.28 g mL (2.28 mmol) of 1,3-propane sultone was added to the reaction mixture until complete dissolution of the initial substances. The reaction was carried out at room temperature for 48 hours in an argon atmosphere. Next, the reaction mixture was precipitated in diethyl ether. After the following decantation, the sediment was dissolved in methanol and concentrated on the rotary evaporator.”

“3.1.4. 4,8,14,18,23,26,28,31,32,35-deca-[thioethane (2'-(N,N - dimethyl, N-acetate)ammonium)ethoxy]-pillar[5]arene (9).

In the round bottom flask equipped with magnetic stirrer, 0.3 g (0.107 mmol) of the macrocycle 5 were dissolved in 10 mL of anhydrous tetrahydrofuran (THF). Then, 0.8 g (2.14 mmol) of lithium hydroxide were dissolved in 2 mL of distilled water and were added to the reaction mixture. The reaction was carried out at room temperature for 24 hours. Next, the precipitate that was formed during the reaction was filtrated, and the filtrate was concentrated on the rotary evaporator.”

Finally, my opinion is that this article has very poor English, very bad presentation of the obtained results some of the results in the text just described those mentioned in the tables, so there is no discussion, very bad described synthetic procedure and it could not be accepted for publication in the current form.

Answer: made by the reviewer were considered. English is edited.

Reviewer 3 Report

Comments and Suggestions for Authors

This manuscript discloses the synthesis of several different pillarenes, characterises them as drugs per se, and explores their antimicrobial properties. A benzenesulfonamide-functionalized pillarene (designated 6) showed the most favorable antimicrobial properties.

I am familiar with pillarene structures, but I think a typical reader of Pharmaceutics may not be. I recommend that the authors show the complete structure for 6, since it is the focus of the work.

In my experience, MICs are expressed either in µM or µg/mL. The authors have used the latter, but report their drug potencies in molar units. My personal preference would be to express concentrations in mM units as well. I think these units will be more familiar to most readers. For example, the MIC for 6 against S. typhimurium is reported as 3.75 x 10^-5. This would be more easily understood, I think, as 37.5 µM. Most of the compounds show MIC values of >300 µM. This is a reasonable cutoff for activity, although a drug having a MIC of >100 µM is unlikely to inspire interest. This does not make the units wrong and I leave to the authors and editor to decide if a change to µM would clarify the text.

The experimental characterisation of the products seems adequate, but there is little information concerning the isolation procedure. For the key compound, 6, the only information recorded in the manuscript is precipitation from ether. Although considerable spectroscopic information is included, I did not find any melting points in the experimental. I did not check the supplementary information. It doesn't matter if it is included there, it should be included in the main text.

When I read the title, I thought that the enhanced activity of sulfonamides would result from some sort of complexation with the pillarenes. It might be useful to consider using some variant of the term sulfonamide-derivatised pillarenes. I think that the work presented is sound although the results seem to me to be modest. In addition to the comments above, I noticed a few other things that should be corrected. These follow.

"the findings of supramolecular chemistry:" findings should be application

BPC should be defined at the first use rather than in Table 3 or the experimental section.

PDI should also be defined and any conclusion drawn from the zeta-potential values should be discussed if it is to be included in Table 1.

ethylidide is misspelled line 82

I don't understand the use of the term propanesulfone (line 89) as it refers to either 7 or 8.

Microorganism is misspelled in Table 3 (line 1580

The number 6 should be bold on line 396.

Comments on the Quality of English Language

Minor editing of English language required

Author Response

Referee: 2

This manuscript discloses the synthesis of several different pillararenes, characterizes them as drugs per se, and explores their antimicrobial properties. A benzenesulfonamide-functionalized pillararene (designated 6) showed the most favorable antimicrobial properties.

  1. I am familiar with pillararene structures, but I think a typical reader of Pharmaceutics may not be. I recommend that the authors show the complete structure for 6, since it is the focus of the work.

Answer: We agree with the reviewer's comment. For better understanding, the complete structure of macrocycle 6 has been added to scheme 1.

Scheme 1. Synthesis of the macrocycles 2-9.

  1. In my experience, MICs are expressed either in µM or µg/mL. The authors have used the latter, but report their drug potencies in molar units. My personal preference would be to express concentrations in mM units as well. I think these units will be more familiar to most readers. For example, the MIC for 6 against S. typhimurium is reported as 3.75 x 10^-5. This would be more easily understood, I think, as 37.5 µM. Most of the compounds show MIC values of >300 µM. This is a reasonable cutoff for activity, although a drug having a MIC of >100 µM is unlikely to inspire interest. This does not make the units wrong and I leave to the authors and editor to decide if a change to µM would clarify the text.

Answer: Thank you for your comment! We agree that expressing MIC in µM is more convenient and more clarify than in M. The necessary substitutions were made throughout the article and in the ESI file.

  1. The experimental characterization of the products seems adequate, but there is little information concerning the isolation procedure. For the key compound, 6, the only information recorded in the manuscript is precipitation from ether. Although considerable spectroscopic information is included, I did not find any melting points in the experimental. I did not check the supplementary information. It doesn't matter if it is included there, it should be included in the main text.

Answer: The main article material now includes information regarding the melting points of newly synthesized compounds 2-9. Furthermore, synthesis methods were revised accordingly. The procedure for selecting target connections has been expanded for 3-7 macrocycles. Lastly, a new fragment has been introduced into the article text:

“3.1.2. General procedure for the synthesis of macrocycles 3-7

In the round bottom flask equipped with a magnetic stirrer, 0.15 g (0.078 mmol) of the macrocycle 2 were dissolved in 7 mL of anhydrous DMF. Then 2.34 mmol (0.15 ml) of methyl iodide/ or 0.19 mL of ethyl iodide/ or 0.26 mL of ethyl 2-bromoacetate/ or 0.64 g of 2-bromo-N-(4-sulfamoylphenyl)acetamide /or 0.41 mL of benzyl bromide were added dropwise to the reaction mixture until complete dissolution of the initial substances. The reaction was carried out at room temperature in for 48 hours an argon atmosphere. Next, the products were isolated by precipitation from diethyl ether. The precipitate that formed was separated by filtration on a filter funnel and washed with diethyl ether.”

When I read the title, I thought that the enhanced activity of sulfonamides would result from some sort of complexation with the pillararenes. It might be useful to consider using some variant of the term sulfonamide-derivatised pillararenes. I think that the work presented is sound although the results seem to me to be modest. In addition to the comments above, I noticed a few other things that should be corrected. These follow.

 "the findings of supramolecular chemistry:" findings should be application

BPC should be defined at the first use rather than in Table 3 or the experimental section.

PDI should also be defined and any conclusion drawn from the zeta-potential values should be discussed if it is to be included in Table 1.

ethylidide is misspelled line 82

I don't understand the use of the term propanesulfone (line 89) as it refers to either 7 or 8.

Microorganism is misspelled in Table 3 (line 1580)

The number 6 should be bold on line 396.

Answer:  All the reviewer's comments have been taken into account. All suggested replacements have been made. Typos corrected: 1,3-propanesultone (line 93), ethyl iodide (line 86), microorganism (in Table 3). Definitions for abbreviations have been added to the text of the article.

Also this fragment:

“Further, it was shown using DLS that pillar[5]arenes 4, 6, 8 do not form stable self-associates in the whole range of concentrations investigated in water. However, macrocycles 3, 5, 9 form stable self-associates in the concentration of 10 µМ, ζ = 18-47 mV (Table 1). This seems to be due to the higher lipophilicity of compounds 3, 5, 9 compared to 4, 6, 8. The analysis of experimental data showed that compounds 4, 6, 8 are in individual form in solution.”

has been replaced in the manuscript on:

“Further, DLS method showed for aqueous solution of pillar[5]arenes 4, 6, 8 no formation of stable self-associates in the whole range of concentrations investigated. Polydispersity index (PDI) values for 4, 6, 8 ranged from 0.38 to 0.62, while ζ - the potential of the system did not exceed 4 mV (Table 1, ESI, Table S1), indicating the absence of stable colloidal systems. However, macrocycles 3, 5, 9 form self-associates in the range of 10-1000 µM concentrations in water. Thus, the PDI for 3, 5, 9 ranged from 0.27 to 0.37 (10 µM) and the ζ-system potential was in the range of 17-47 mV (Table 1, ESI, Table S1). Such high ζ-potential values indicate the formation of a stable colloidal system. It is evident that the aggregation properties of macrocycles in this series of pillar[5]arenes (3-6, 8 and 9) are influenced by the nature of the substitute in the ammonium nitrogen atom and the nature of the anion. Thus, analysis of experimental data showed that compounds 4, 6, 8 are present in the solution individually.”

Round 2

Reviewer 1 Report

Comments and Suggestions for Authors

My minor comments have been adressed by the Authors of this manuscript. On the other hand, I still find the data preliminary and in my opinion the study is limited in its scope. However, it may be well suited to the topic of the special issue.

Comments on the Quality of English Language

The usage of English is fair.

Reviewer 2 Report

Comments and Suggestions for Authors

No more comments. The paper now can be accepted for publication. 

Reviewer 3 Report

Comments and Suggestions for Authors

The authors considered my suggestions and it appeared to me that their alterations were appropriate. My only issue was that some of the revised text was less grammatical than in the initial manuscript.

Comments on the Quality of English Language

My only issue was that some of the revised text was less grammatical than in the initial manuscript.